# Current Evolutionary Dynamics of Porcine Epidemic Diarrhea Virus (PEDV) in the U.S. a Decade After Introduction

**DOI:** 10.3390/v17050654

**Published:** 2025-04-30

**Authors:** Joao P. Herrera da Silva, Nakarin Pamornchainavakul, Mariana Kikuti, Xiaomei Yue, Cesar A. Corzo, Kimberly VanderWaal

**Affiliations:** Department of Veterinary Population Medicine, College of Veterinary Medicine, University of Minnesota, St. Paul, MN 55108, USA; pamor001@umn.edu (N.P.); mkikuti@umn.edu (M.K.); yue00075@umn.edu (X.Y.); corzo@umn.edu (C.A.C.)

**Keywords:** PEDV, coronavirus, recombination, genetic diversity, temporal dynamics

## Abstract

Porcine Epidemic Diarrhea Virus (PEDV) was introduced in the United States (U.S.) in 2013, spreading rapidly and leading to economic losses. Two strains, S-INDEL and non-S-INDEL, are present in the U.S. We analyzed 313 genomes and 556 Spike protein sequences generated since its introduction. PEDV case numbers were highest during the first two years after its introduction (epidemic phase), then declined and stabilized in the following years (endemic phase). Sequence surveillance was higher during the initial epidemic phase. Our results suggest the non-S-INDEL strain is the predominant strain in U.S. The non-S-INDEL sequences exhibit pairwise nucleotide identity percentages above 97.6%. Most non-S-INDEL sequences sampled after 2017 clustered into two sub-clades. No descendants derived from other clades present in the epidemic period were detected in the contemporary data, suggesting that these clades are no longer circulating in the U.S. The two clades currently circulating are restricted to two respective geographic regions and our results suggest limited inter-regional spread. This insight helps determine the risk of re-introduction of PEDV if it were regionally eliminated. Ongoing molecular surveillance is essential to confirming that some older clades no longer circulate anymore in the U.S., mapping the distribution and spread of recent clades, and understanding PEDV’s evolutionary diversification.

## 1. Introduction

Porcine Epidemic Diarrhea Virus (PEDV) causes acute enteric disease in pigs, which is often lethal in suckling piglets [1,2,3]. PEDV is widely dispersed throughout the world and is responsible for significant losses in the swine industry globally [4,5,6,7]. PEDV was first reported in the United States (U.S.) around April of 2013, and was responsible for a major epidemic in the country [6,7,8]. After its introduction, the virus spread rapidly across the U.S. and has been circulating ever since in swine herds. The epidemic phase of PEDV in the U.S., a period characterized by rapid between-farm spread and high incidence, lasted from April 2013 to August 2014. This has been followed by an endemic phase in which disease incidence stabilized up to the present [9,10]. The primary mode of transmission for PEDV is through the fecal–oral route [11]. The following activities have been reported as main modes of PEDV spread between farms: transportation of infected animals from one farm to another, fomites carried by movement of farm employees or vehicles between farms, and contaminated feed, among others [12,13,14,15].

The causative agent of this disease, *Alphacoronavirus porci*, is an enveloped, positive-sense, single-stranded RNA virus of the *Coronaviridae* family, *Alphacoronavirus* genus, with a genome of approximately 28 kb [16]. Its genome consists of ORF1a and ORF1b, encoding 16 non-structural proteins involved in replication, transcription, and immune evasion, and four other ORFs that encode for structural protein: the spike protein (S), nucleocapsid (N), membrane (M), envelope (E), as well as the accessory protein ORF3. RNA viruses are characterized by rapid evolutionary dynamics, which allows them to adapt quickly to the niches they occupy [17,18]. Most RNA viruses have high mutation rates due to the absence of proofreading activity in their replicases [19,20]. Coronaviruses encode a proofreading exonuclease (nsp14), causing these viruses to exhibit lower mutation rates compared to other RNA viruses [21,22]. Recombination is another mechanism that drives the diversification of viral populations. Recombination enables the emergence of new haplotypes, enables new epistatic interactions, and can purge deleterious mutations [20,23,24,25]. Coronaviruses exhibit high rates of recombination, and recombination events have been reported in all seven ORFs of PEDV [26,27,28,29].

PEDV is divided into two major genogroups based on the phylogeny of the spike protein, and the G1 and G2 genogroups are further subdivided into five subgroups: G1a/b, and G2a/b/c [6,30,31]. In the U.S., the two strains present are the S-INDEL strain (low pathogenic), which is phylogenetically related to genogroup G1b, and the non-S-INDEL strain (high pathogenic), which is related to genogroup G2b. These two strains are characterized by the presence or absence of deletions in the ORF-S gene, respectively [3,8,32,33]. The PEDV non-S-INDEL strain was first detected in April 2013, whereas the S-INDEL strain was reported in June 2013 [6]. Phylogenetic analyses demonstrated that PEDV sequences in the U.S. were closely related to PEDV from China, suggesting a possible origin of PEDV found in the U.S. [3,6]. Recent studies using phylogeographic approaches have highlighted connectivity between several swine-producing countries, allowing the identification of key contributors to the transboundary transmission chain [3,34,35,36]. For example, several “US-like” PEDV strains have been documented in relation to outbreaks in other countries, suggesting that the United States has played a role in the spread of PEDV to countries, including South Korea, Japan, China, and Mexico [3,34,35,36].

Most studies investigating the evolutionary dynamics of PEDV in the U.S. were carried out only at the early stages of the epidemic, shortly after its introduction [3,6,8,29,32]. The evolutionary dynamics of PEDV during the endemic period in the country remains unexplored. Here, we aim to investigate the temporal dynamics of PEDV in the U.S. over more than a decade since its introduction. By estimating the genetic diversity and variability of PEDV and through Bayesian inference, we characterized the changes in the genetic composition of the PEDV population over time.

## 2. Materials and Methods

### 2.1. Data Source

A dataset consisting of 556 PEDV spike protein sequences from the U.S., generated between 2013 and 2024, was assembled. Of these, 228 sequences were downloaded from GenBank [37] and 328 were obtained from the Morrison Swine Health Monitoring Project (MSHMP) [38]. MSHMP is a voluntary disease monitoring program that aggregates industry data on swine disease occurrence, accounting for approximately 60% of the breeding herd population [9,38]. The project also curates PEDV sequences generated by participants through their routine monitoring efforts in breeding, gilt developing units, growing and finishing herds. In addition, a second dataset comprising 313 PEDV full-length genomes from the U.S., generated between 2013 and 2017 was constructed, which included 73 sequences obtained from MSHMP and 240 sequences retrieved from GenBank. The sequences included in this study originated from 21 states that include all major swine producing regions. For both complete genomes and spike protein, we used all available sequences from both databases.

For the purpose of this study, the “epidemic period” was defined as spanning from May 2013 to August 2014, during which disease prevalence peaked at 31.6% of monitored sow farms [9,10]. The “endemic period” was defined as beginning in August 2014 and continuing thereafter, during which the prevalence decreased and stabilized, reaching a peak of 5.5% in 202 [9,10].

### 2.2. Sequence Alignment

Using the default parameters, sequence alignment was performed using MAFFT V7.490 [39]. The sequences were trimmed by removing the 5′ and 3′ UTR regions and only the coding regions of the genomes were used in our analyses. Sequences were classified as the S-INDEL or non-S-INDEL strain based on the presence of insertions and deletions at specific positions in the coding region of the Spike protein, as described by Wang et al. [8].

### 2.3. Sequences Comparison

Genetic distance for both the complete genome and the spike protein within and between clades was estimated as the pairwise nucleotide identity percentage using sequence tool demarcation SDT v.1.2 [40]. Pairwise identity percentages were estimated for each strain, between strains, and between the current clades and the ancestral (epidemic period) clades of the non-S-INDEL strain.

### 2.4. Recombination Analysis

Complete PEDV genomes and spike (S) protein sequences were scanned to infer putative recombination events using the Rdp [41], Geneconv [42], Bootscan [43], Max χ2 [44], Chimaera [45], Siscan [46], and 3Seq [47] methods implemented in RDP5 [48] using default parameters. Only recombination events supported by at least 4 different methods and *p*-values lower than α = 0.05 (Bonferroni corrected) were considered to be reliable. Phylogenetic networks were inferred to capture patterns of non-tree-like evolution using the Neighbor-Net algorithm implemented in SplitsTree v. 4.19.2 [49]. Distances were calculated using the GTR+G substitution model determined in jModelTest2, using 1000 bootstrap replicates.

### 2.5. Phylogenetic Analysis

Phylogenetic trees were inferred by maximum-likelihood implemented in RAxML-NG [50]. The nucleotide substitution model was determined using ModelTest-NG [51]. To assess whether the data exhibited evidence of temporal signal, TempEst was used to perform a root-to-tip regression on ML phylogenetic trees [52]. TempEst is useful for exploratory analyses of temporal signal and identifying outliers or sequences with incorrect dating [53,54]. However, several limitations have been discussed regarding the use of root-to-tip regression as a formal statistical test, such as the lack of data independence and the assumption of strict molecular clock-like behavior. Therefore, we employed the Bayesian Time Signal (BETS) algorithm [53,54], implemented in BEAST V.1.10.4 [55], as a formal test to assess the temporal signal structure of PEDV populations. The BETS analysis is based on the comparison of the fit of competing models, one that incorporates time and that does not, which are distinguished by the Bayes factor. To determine the evidence of a temporal signal, we performed simulations where we provided sequence date information (heterochronous date) and where no date information was provided (isochronous date). The simulations were performed using both a strict molecular clock and a relaxed uncorrelated lognormal molecular clock. The performance of each simulation was evaluated by estimating the (log) Bayes factor [56], which was estimated based on the difference in the (log) maximum marginal likelihoods of the competing models. The marginal (log) likelihoods were estimated by generalized stepping-stone (GSS) sampling [57]. We used an MCMC chain length of 1 × 10^8^ with 25% burin and 100 stepping stones, with a sub-chain MCMC length of 1 × 10^6^. The convergence of the analyses was assessed using Tracer v1.7 and was determined based on effective sample size of at least 200, and by the degree of interdependence of the samples assessed based on the degree of mixing of the parameters [58].

### 2.6. Genetic Diversity

The average nucleotide diversity index π [59], was computed through pairwise sequence comparisons using a Python v3.8.11 script [60] for both complete genomes and individual coding regions. We estimated the 95% confidence intervals for the mean π values using a bootstrap test with 1000 non-parametric simulations [60], performed with the Simpleboot package in R software [61]. This approach was used to assess the statistical significance among the π values. π values per site were estimated for each ORF individually and for the whole genome set, as well as for each strain individually (PEDV S-INDEL and PEDV non-S-INDEL). In order to identify the genomic regions exhibiting the highest levels of genetic variability, the nucleotide diversity per site across the genome was calculated using a sliding window of 100 nucleotides with a step size of 10 nucleotides, utilizing DnaSp v.5.10 [62].

### 2.7. Selections Analysis

Selection analyses were performed to infer how natural selection acts on different ORFs encoding structural proteins of the PEDV genome. Recombination events can confound the search for sites under selection, so to avoid noise generated by recombination, a search for recombination breakpoints was performed for each ORF using Genetic Algorithm Recombination Detection (GARD) [63]. We then performed a search for potential sites under positive or negative selection in each of the ORFs encoding structural proteins. Three different methods were used to screen sites for evidence of selection: Single Likelihood Ancestor Counting (SLAC) [64], Mixed Effects Model of Evolution (MEME) [65], and Fast Unconstrained Bayesian Approximation (FUBAR) [66], all implemented in the DataMonkey web server [67]. We also estimated the ratios between synonymous and nonsynonymous substitution rates (ω) using SLAC [64].

## 3. Results

### 3.1. Stabilization of PEDV Case Numbers After the Epidemic Period

The occurrence of PEDV in the country was evaluated based on the number of cases reported by production systems participating in MSHMP. Here, “case numbers” refer to the number of sow farms reporting an outbreak. Following the introduction of PEDV, a massive number of cases were reported, especially during the first two years (epidemic period). After this initial surge, the number of cases decreased and became stable, with slight fluctuations, as expected in an endemic period. However, 2022 was an unusual year, showing a slight increase in the number of cases (Figure 1).

In total, we analyzed 556 spike protein sequences, with 499 corresponding to the non-S-INDEL strain and 57 to the S-INDEL strain. Additionally, we analyzed 313 full genomes, of which 288 corresponded to the non-S-INDEL strain and 25 to the S-INDEL strain. These findings suggest that the non-INDEL strain is the predominant strain in the U.S.

PEDV sequence monitoring was intense during the epidemic period. Afterwards, the number of sequences obtained decreased, particularly after 2019, when fewer than ten sequences per year were generated. In terms of sequencing effort, the ratio of sequences per number of infected sow farms was fairly constant at a median of 2.2 sequences per ten infected sow farms across 2013–2024 (with an interquartile range of 1.3 to 3.7). At least one sequence per ten infected sow farms was obtained, except in 2021 and 2022, when only 0.74 and 0.49 sequences were generated, respectively (Figure 1). It is important to note that the case data available includes only those reported in breeding herds and exclusively those submitted to the MSHMP. As a result, the actual number of cases may be higher than what is presented here. Furthermore, the sequences we analyzed do not necessarily originate solely from breeding herds as some may have been obtained from grow–finish farms.

### 3.2. Evidence of Multiple Recombination Events

Recombination analyses were conducted to understand the contribution of horizontal transfer of genetic material to the diversification of PEDV populations in the U.S. In total, we identified 12 recombination events across the full genome (Table 1A) that were detected by at least four different methods implemented in RDP5 with a *p*-value less than 0.05.

Among these events, four warrant special attention due to their detection in a large number of sequences. Event 6 in Table 1A involved a recombinant block located in the ORF1b region, more specifically, spanning nsp13 and nsp15. This event was detected in 106 PEDV genomes (Table 1A). Event 10 was detected in 116 sequences, with the recombination breakpoint located between nsp4 in ORF1a and nsp12 in ORF1b (Table 1A). Event 11 was detected in 223 sequences, with the recombination breakpoint located between nsp3 in ORF1a and nsp12 in ORF 1b (Table 1A). Event 12 was detected in 152 sequences, with the recombination breakpoint located between nsp15 in ORF1b and the Spike protein (Table 1A).

Recombination analyses were also performed specifically for the S1 subunit of the Spike protein. Only two well-supported recombination events were identified in the S1 region (Table 1B). These results differ somewhat from the full-genome analysis, where we were able to identify a greater number of recombination events in the Spike protein. Some sequences that showed evidence of recombination in the spike protein region in the full-genome analysis were not detected as recombinants when analyzing only the Spike protein dataset. In the full-genome analysis, 179 sequences were identified as recombinants, with breakpoints covering the S1 region of the Spike protein (positions 20,342 to 22,562). However, when analyzing only the S1 region dataset (*n* = 566 sequences), only 61 sequences were detected as recombinants. We hypothesize that the loss of recombination signal in these datasets may be because this analysis was performed using partial sequences. Partial sequences lack the context of the complete genome, weakening the sensitivity of the analysis. The majority of sequences that exhibited recombination events in the spike protein region were detected in samples collected before 2017. Only four sequences sampled from 2018 onward showed recombination events in the spike protein. However, we cannot rule out the possibility that some of these more recent sequences may also contain undetectable recombination events using RDP, given that they are not full genomes. This underscores the importance of complete genomes in obtaining a more comprehensive and accurate understanding of the underlying processes driving viral diversification.

Bifurcating phylogenetic trees are not always capable of capturing the evolutionary history of organisms, especially those that experience frequent recombination events [49]. To assess the impact of recombination events in shaping the genealogical trajectory of complete PEDV genomes, we reconstructed phylogenetic networks using SplitsTree4 [49] to capture the non-tree-like evolutionary patterns, which could potentially be explained by evidence of recombination events. Based on the topology of the phylogenetic networks, the PEDV full genome were grouped into four clusters (Figure 2): the cluster that includes the S-INDEL strain shows a long branch, reflecting the high divergence between these S-INDEL and non-S-INDEL sequences, and three other genetically more similar clusters comprising sequences of the PEDV non-S-INDEL strain (Figure 2). The lateral edges (reticulations) connecting the branches of the phylogenetic network provide evidence of lateral transfer of genetic material between clusters (Figure 2). Additionally, we performed the PHI test, which detected significant evidence of recombination events among the PEDV full genome sequences (*p* < 0.001).

Our phylogenetic networks based on the S1 region of the Spike protein displayed a topology similar to that observed in the networks reconstructed for complete genomes. A long branch separating the S-INDEL from the non-S-INDEL strains indicates a high level of divergence between them. The high degree of reticulation was observed in the backbone of the phylogenetic network, particularly in the region where the S-INDEL and non-S-INDEL strains diverge, suggesting occurrences of inter-strain recombination inter-strain. A lower degree of reticulation was observed along the internal branches of the S-INDEL strain, suggesting that intra-strain recombination for S-INDEL was less frequent (Figure 3). The PEDV non-S-INDEL cluster was subdivided into four groups, and a high degree of reticulation was observed, suggesting that intra-strain recombination events were more frequent among non-INDEL strains (Figure 3). The recombination evidence was confirmed by the PHI test (*p* < 0.001).

### 3.3. Temporal Signal and Nucleotide Substitution Rates

Phylogenetic trees based on the full genomes and the Spike protein were reconstructed using maximum likelihood in RAxML-NG [50], employing the GTR+I+G4 nucleotide substitution model, which was determined to be the best fit according to the Akaike Information Criterion (AIC). The phylogenetic tree reconstructed for complete genomes displayed a topology with a more basal group composed of sequences belonging to the PEDV S-INDEL strain and another clade containing the PEDV non-S-INDEL strain (Figure 4). The non-S-INDEL clade can be subdivided into two clusters (Figure 4). Due to the numerous recombination events throughout the complete genome, distributed across various regions, we were unable to perform time-scaled tree analyses for whole genomes. The occurrence of recombination events violates the assumptions of this analysis and affects coalescence times as well as evolutionary rates.

Recombination events were identified in the spike protein region, as previously mentioned. For subsequent analyses, all sequences presenting evidence of recombination were excluded, including those identified solely within the complete genome dataset as well as the Spike protein analysis. A particularly conservative approach was adopted, whereby any sequence showing recombination evidence was removed, irrespective of the number of detection methods that confirmed the events, resulting in a dataset of 320 sequences from the non-S-INDEL strain and 43 sequences from the S-INDEL strain. As mentioned in the previous section, only four of the most recent sequences showed evidence of recombination, while the others were all generated before 2017.

Maximum-likelihood trees were used to conduct root-to-tip regression analyses in TempEst [52] as an exploratory approach to assess evidence of temporal signal. Both strains demonstrated reasonable evidence of a temporal signal: the PEDV S-INDEL strain showed a correlation of 0.769 with an R^2^ value of 0.593, while the PEDV non-S-INDEL strain exhibited a correlation of 0.785 and an R^2^ value of 0.594. To formally assess the temporal signal, we conducted a Bayesian Evaluation of Temporal Signal (BETS) [53,54]. In all BETS analyses, the (log) Bayes Factor exceeded five, favoring the heterochronous model over the isochronous model and providing significant evidence for a temporal signal in both strains (Appendix A). Based on the log Bayes factor, the strict molecular clock was the best-fit for the PEDV S-INDEL strain, while the uncorrelated relaxed clock lognormal model was preferred for the PEDV non-S-INDEL strain.

The mean nucleotide substitution rate was estimated at 7.01 × 10^−4^ subs/site/year (95% HPD: 3.27 × 10^−4^ to 1.16 × 10^−3^) for the PEDV S-INDEL strain spike protein and 1.41 × 10^−3^ subs/site/year (95% HPD: 1.19 × 10^−3^ to 1.65 × 10^−3^) for the PEDV non-S-INDEL strain. The PEDV S-INDEL strain has a TMRCA around 1991 (95% HPD: 1955.0 to 2012.6), while the PEDV non-S-INDEL strain presented a TMRCA around 2010 (95% HPD: 2006.9 to 2012.2) (Appendix A).

Interestingly, most sequences of the PEDV non-S-INDEL strain sampled after 2017 clustered into two monophyletic clades (Figure 5). Clade 1 has a TMRCA estimated at 2017.2 (95% HPD: 2017.4 to 2018.7), while clade 2 has a TMRCA estimated at 2016.0 (95% HPD: 2015.1 to 2016.6). We detected few descendants from other clades that were circulating during the epidemic period. Only singletons and another small (clade 3) were identified (Figure 5), but after 2021, they were no longer detected.

### 3.4. Genetic Variability in U.S. Population

Genetic variability was assessed for the complete set of sequences and for each strain individually. Nucleotide diversity was estimated for complete genomes. We also estimated the genetic diversity for each individual coding regions to evaluate the relative contribution of the genetic diversity of each gene to the diversity observed in the complete genome (Figure 6). When analyzing the complete dataset (S-INDEL and non-S-INDEL combined), we found that the Spike (S) protein exhibited the highest nucleotide diversity by far (0.015), followed by the Envelope (E) (0.0038), ORF1b (0.0033), ORF1a (0.0026), Nucleocapsid (N) (0.0019), ORF3 (0.0016), and finally, the Matrix (M) protein (0.0010). All showed substantial differences in their mean π values (Figure 6).

The PEDV non-S-INDEL strain exhibited the highest nucleotide diversity in the Spike (S) protein (0.0054), followed by the Envelope (E) protein (0.0036), ORF1a (0.0023), ORF1b (0.0017), ORF3 (0.0014), Nucleocapsid (N) (0.0014), and finally, the Matrix (M) protein (0.0007). All of these differences were statistically significant (Figure 6).

Although the number of sequences for the S-INDEL strain was low, it exhibited significantly higher nucleotide diversity compared to the non-S-INDEL strain, both at the full genome level and for each individual ORF analyzed (Figure 6). For the PEDV S-INDEL strain, the highest nucleotide diversity was observed in the ORF1b (0.009) and Nucleocapsid (N) (0.008) regions (Figure 6), with no statistically significant differences between them. These were followed by ORF1a (0.006), Envelope (E) (0.006), and Spike (S) (0.006), which also showed no statistically significant differences and overlapping confidence intervals with ORF N. The lowest nucleotide diversity was observed in the Matrix (M) (0.004) and ORF3 (0.003) regions (Figure 6), with no significant differences between these two genes either (Figure 6).

The differences in nucleotide diversity between genes indicate that each region of the genome exhibits with some genomic regions being more prone to accumulating genetic variation than others. The hypervariable regions in ORF-S and ORF-1b appear to coincide with regions where recombination has been detected, suggesting a possible explanation for the high genetic variation. We also observed differences between the two strains regarding which genes show greater genetic diversity, suggesting distinct evolutionary trajectories between the two PEDV strains.

To better understand how genetic diversity is distributed across the PEDV genome, we measured nucleotide diversity by site using a sliding window approach (100 nt window with 25 nt steps). The analyses were performed on the complete set of sequences as well as for each strain separately. We identified several peaks of nucleotide diversity that exceeded the mean across the genome (Figure 7). Overall, the distribution of nucleotide diversity along the genome was similar between the complete dataset and the PEDV non-S-INDEL strain. Both exhibited hypervariable regions in the 5′ end of the genome, specifically within the nsp2 and nsp3 regions. Another hypervariable region was located in the S1 subunit of the Spike (S) protein, particularly in the C-domain (Figure 7). The dataset combining both PEDV S-INDEL and PEDV non-S-INDEL strains revealed an additional hypervariable region in the NTD of the S1 subunit of the Spike protein. This pattern was not observed in the PEDV non-S-INDEL strain, suggesting that hypervariable regions may vary by strain, reflecting distinct evolutionary trajectories between them (Figure 7).

The PEDV S-INDEL strain also displayed a hypervariable region within the nsp2 and nsp3 regions, as well as in ORF1b, particularly in the nsp12, nsp14, nsp15, and nsp16 regions. In contrast, the hypervariable regions in the S1 subunit of the Spike protein observed in the PEDV non-S-INDEL strain were not detected in the S-INDEL strain (Figure 7).

### 3.5. High Sequence Similarity Between PEDV Strains and Among the Current Clades

Complete genome analyses revealed a high percentage of pairwise nucleotide identity for the two PEDV strains, above 98% for both intra-strain and inter-strain comparisons (Table 2). Sequence comparisons were also performed for datasets of the Spike protein. A high pairwise identity percentage was observed within the non-S-INDEL strain, with a mean of 99.1% (Table 2). We computed the percentage of pairwise nucleotide identity within and between the two current clades, as well as between the clades that circulated before 2017. The average pairwise nucleotide identity within the new clades was 99.3% for clade 1 and 99.2% for clade 2 (Table 2). However, comparisons between clades 1 and 2, as well as between these clades and those that circulated before 2017, showed average pairwise nucleotide identity values ranging between 98.6 and 99.3%, respectively (Table 2). The S-INDEL strain exhibited intra-strain pairwise nucleotide identity above 94.5%, with an average pairwise identity of 99.1%. Pairwise nucleotide identity between the S-INDEL and non-S-INDEL strains ranged from 91.1% to 94.3% (Table 2). These results indicate that the sequences within each clade are genetically very similar. However, there is a greater genetic distance between the S-INDEL and non-S-INDEL clades in the Spike protein.

### 3.6. Positive Selection Evidence in a Spike Protein Epitope

Selection analyses were conducted to infer how natural selection modulates variation in regions encoding structural proteins, which are the main targets of the immune system [68]. For this, we estimated the rates of synonymous and non-synonymous substitutions (dN/dS) for the ORFs S, E, M, and N. For both PEDV S-INDEL and PEDV non-S-INDEL strains, the dN/dS values were less than one, highlighting the action of purifying selection on these genes (Table 3). The dN/dS ratios varied across the genome, with purifying selection acting more strictly on ORF M and more relaxed on ORFs S, E, and N, respectively. Although the dN/dS values were lower for the PEDV S-INDEL strain, the small sample size could have affected this estimation (Table 3).

The spike protein exhibited the highest number of sites under both purifying and diversifying selection, followed by the nucleocapsid protein. The matrix and envelope proteins exhibited very few sites under selection (Table 3). This pattern was observed for both strains. Of all the sites under diversifying selection, only one was detected in a known epitope of the spike protein (position 723), resulting in a serine-to-asparagine substitution.

## 4. Discussion

The introduction of PEDV into the U.S. caused significant economic losses to the U.S. swine industry and affected pork availability in the market [12,69,70]. The high transmission rate, combined with the movement of animals within the country, enabled the virus to spread rapidly nationwide [71]. Studies identifying factors that increase the risk of outbreaks have contributed to improving biosecurity protocols guiding animal management strategies [9,72,73,74]. The reduction in breeding herd incidence reflects the results of these efforts [9]; however, PEDV continues to circulate in swine herds in the U.S., particularly during the colder seasons of the year [7]. Here, we reconstructed the evolutionary trajectory of PEDV over the 10 years following its introduction.

Our sequence dataset covers 21 different states, encompassing four major swine-producing regions: the Northeast, Midwest, Great Plains, and Atlantic Seaboard. Approximately 263 sequences lacked information regarding their geographic origin. Overall, the temporal distribution of sequences from each of the four regions was relatively similar, except for the epidemic period, during which some areas contributed a higher number of sequences. However, we were not able to assess if sequencing effort was equal across production systems within regions, which could mean that some genetic diversity was missed.

Our results suggest that the PEDV non-S-INDEL is the predominant strain in the U.S., although it is important to consider the potential bias in our dataset. There is a substantial gap in sequence surveillance during the post-epidemic period since sequencing efforts declined drastically after 2015. Although there is no evidence to support this, the possibility that PEDV S-INDEL is silently circulating within livestock populations cannot be ruled out, given that the PEDV S-INDEL strain is less pathogenic. Previous studies show that the non-S-INDEL PEDV strain has much higher transmission rates compared to the S-INDEL PEDV strain, which may contribute to our finding that the S-INDEL strain is less widespread [71].

PEDV populations continue to differentiate over time. Even though the PEDV Spike protein exhibits low intra-strain genetic variability, strong evidence of temporal diversification was detected. Similar patterns have also been observed in SARS-CoV-2 populations [54]. Our phylogenetic analyses based on the S1 subunit of the Spike protein suggest that many of the early clades circulating during the epidemic period are potentially no longer be circulating in the U.S. at present. Many of the older clades appear to have left no descendants. Based on the available data, nearly all sequences sampled after 2017 clustered into two current clades. The persistence of these new clades could be explained by differential adaptability or by successive population bottlenecks and expansions, typical of seasonal respiratory diseases, which randomly contribute to the fixation of certain genotypes and the loss of others within the population [75,76]. Information regarding the pathogenicity and transmissibility of the PEDV genotypes currently circulating in the United States is yet to be uncovered, but deserves further investigation.

Interestingly, the circulation of these two contemporary clades is restricted to two separate geographic regions, suggesting compartmentalized circulation within those regions with limited spread between different regions during the recent endemic period. This pattern seems plausible, as one of the main factors associated with virus dispersal in swine in the U.S. is animal movement between regions [77]. The observation of compartmentalized spread within geographic regions suggests that if PEDV were to be regionally eliminated, then the risk of re-introduction is relatively small. This is particularly relevant given ongoing discussions of eliminating PEDV from the U.S. swine [78].

We identified several hypervariable regions throughout the genome of the U.S. PEDV population. When analyzing each strain separately, we observed intricate patterns of hypervariability specific to each strain. The first hypervariable region is located in ORF1a, identified in both strains, spanning the carboxy-terminal region of nsp2 and the amino-terminal region of nsp3. The high variability in this region can be explained by evidence of recombination events in this portion of the genome. Multiple recombination breakpoints were detected in this genomic region. Another hypervariable region was detected exclusively in the non-S-INDEL strain and is located in the S1 subunit of the Spike protein. A similar pattern of genetic variation in this region, also associated with recombination events, has been previously reported [79,80]. The S-INDEL strain, on the other hand, exhibited a hypervariable region in the ORF1b region. The high nucleotide diversity in this region is associated with evidence of recombination in that region (Figure 7 and Table 1A).

Recombination may play a central role in the diversification of PEDV. Recombination events located in regions associated with immune response deserve special attention, as they may result in the formation of new epitope profiles, leading to immune escape, or potentially contributing to increased pathogenicity and transmission rates, as well as threatening the durability of the vaccines available on the market and vaccine development, as well [79,80,81,82,83,84]. The complete genome analysis revealed some recombination events that were not detected in the Spike protein-only analysis, highlighting the power of complete genomes to more accurately capture recombination events.

Several sites of positive selection were identified in the ORF S region; however, only one of these sites is located within a B-cell epitope (SE16, 722SSTFNSTREL731), warranting special attention as this epitope has been described as having a high antigenic index as well as hydrophilicity, making it likely to interact with the host’s immune system [85]. Furthermore, the relevance of the N723S amino acid substitution extends as this is a potential region for vaccine development [85], and amino acid changes in this region may pose a direct challenge to vaccine stability [86].

A limitation of our findings is that PEDV sequence data are not routinely generated, and thus, we do not have a full picture of PEDV diversity across the U.S. While sequencing effort during the endemic period has been relatively constant (e.g., 1–3 sequences available per 10 reported sow farm outbreaks), taking into account the fact that nearly half of the sequences analyzed in this study lacked information about their geographic origin, it is not clear whether sequencing efforts are proportional to regional incidence patterns [9], or whether sequencing is unevenly pursued across regions or in different production systems. Continued sequence surveillance is vital for the swine industry to advance toward disease eradication and is key to (a) confirming the extinction of older clades, (b) mapping the distribution of recent clades, and (c) understanding PEDV’s evolutionary diversification.

## 5. Conclusions

Our results provide an updated perspective on the evolutionary state of PEDV in the United States over the past decade. The PEDV strains currently circulating in the U.S. are slightly distinct from those present during the epidemic period. Several recombination events have been detected, some of which are widely fixed within North American populations, suggesting an adaptive advantage. The two recent clades reported here are geographically restricted to specific regions, indicating limited epidemiological connectivity leading to exchange of PEDV viruses between these areas. However, the relatively small number of available sequences post-epidemic may represent a potential bias in our dataset. It is crucial to resume more systematic PEDV sequence surveillance, particularly to generate additional complete genomes. This will provide a clearer understanding of the distribution of new clades across different regions and confirm if some of the older clades that circulated in the U.S. during the epidemic period have truly ceased to circulate. Such information is vital to achieving the eradication of PEDV in the United States.

## Figures and Tables

**Figure 1 viruses-17-00654-f001:**
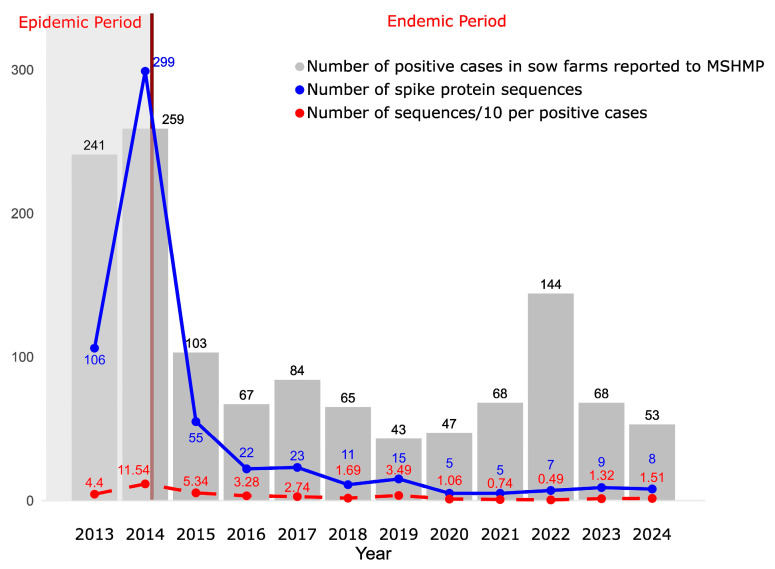
The gray bars represent the number of positive cases in sow farms per year reported to MSHMP. The solid blue line represents the number of spike protein sequences available. The dashed red line represents the number of sequences per 10 sow farms reporting outbreaks. The brown line marks the boundary between the epidemic and endemic periods.

**Figure 2 viruses-17-00654-f002:**
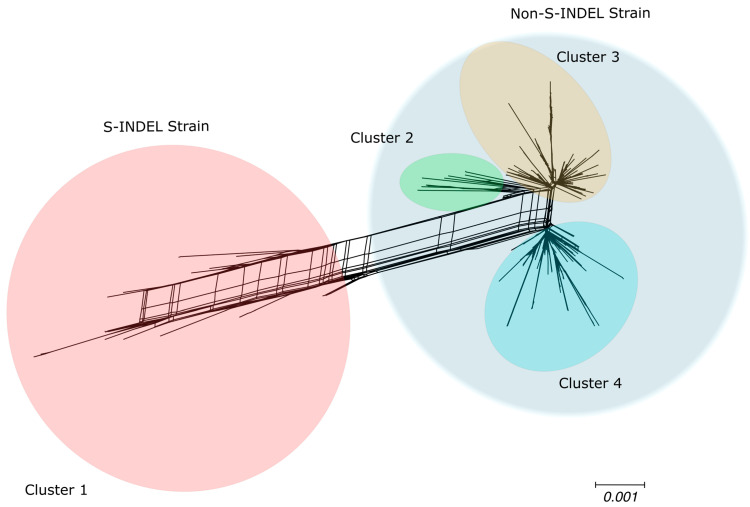
Phylogenetic evidence of recombination between PEDV full genome sequences. Neighbor-Net network analysis was performed using SplitsTree4. The formation of a reticulated network rather than a strictly bifurcating tree suggests evidence of recombination. Large circles highlight the PEDV S-INDEL strain in red and the PEDV non-S-INDEL strain in gray. PEDV non-S-IDEL sequences are further divided into three smaller clusters: cluster 2 in green, cluster 3 in orange, and cluster 4 in blue.

**Figure 3 viruses-17-00654-f003:**
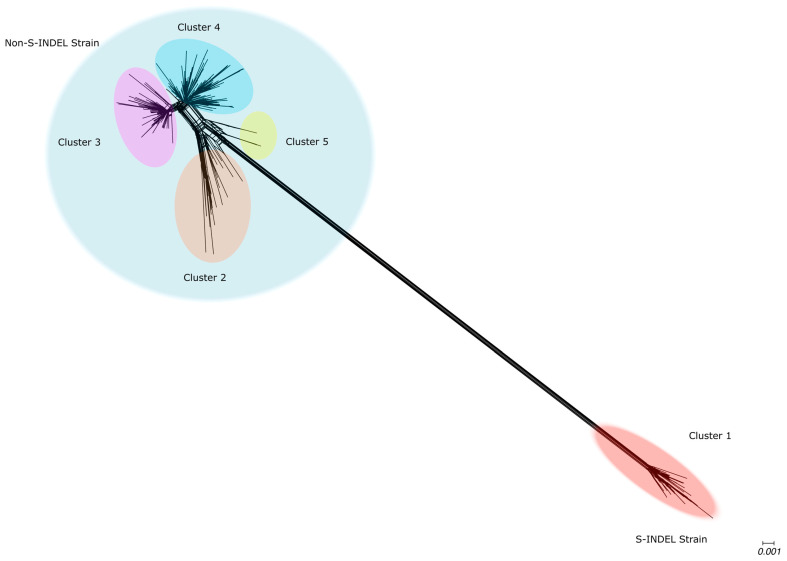
Phylogenetic evidence of recombination between PEDV Spike protein. Neighbor-Net network analysis was performed using SplitsTree4. The formation of a reticulated network, rather than a strictly bifurcating tree, suggests evidence of recombination. Large circles highlight the PEDV S-INDEL strain in red and the PEDV non-S-INDEL strain in gray. PEDV non-S-INDEL sequences are further divided into four smaller clusters: cluster 2 in orange, cluster 3 in pink, cluster 4 in blue, and cluster 5 in yellow.

**Figure 4 viruses-17-00654-f004:**
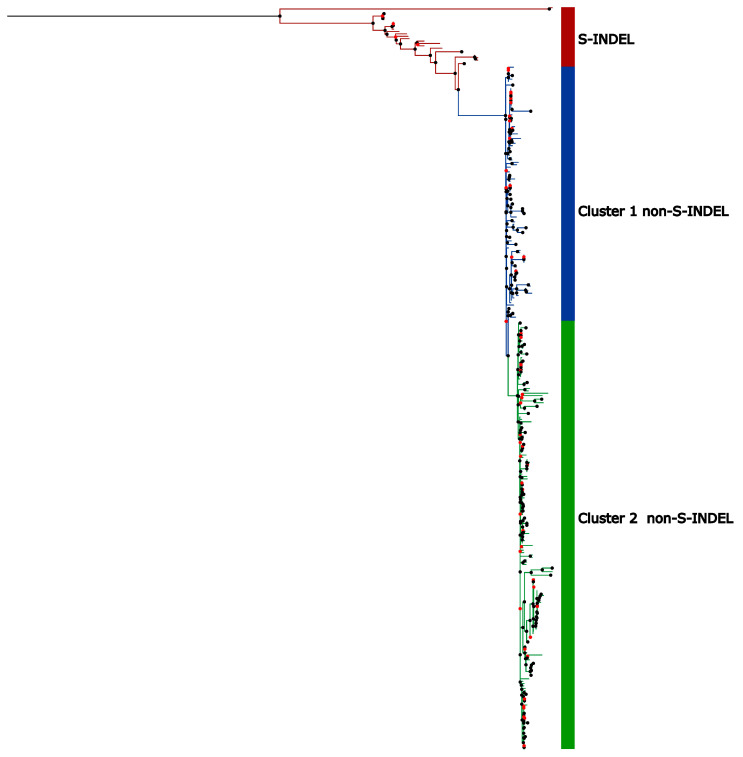
Maximal likelihood phylogenetic tree based on the full genome nucleotide sequences of PEDV. The colored circles on the nodes represent bootstrap support values for the branches: black circles indicate support values greater than 75%, while red circles represent branches with support values below 75%. Sequences belonging to PEDV S-INDEL strain are indicated by the red bar. The PEDV non-S-INDEL strains are divided into two clusters shown on the colored bar: cluster 1 in blue and cluster 2 in green.

**Figure 5 viruses-17-00654-f005:**
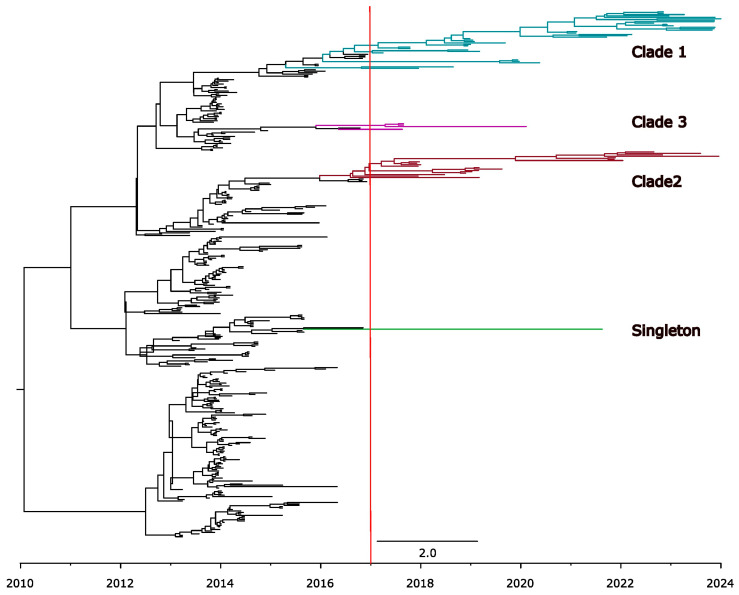
Time-scaled tree of PEDV spike protein sequences from PEDV non-S-INDEL strains. The scale at the bottom of the tree represents the time-scale in years. The earlier clades from the epidemic-period are colored in black, while the clades currently circulating in the U.S. are highlighted in different colors: clade 1 in blue, clade 2 in red, clade 3 in pink, and singleton 1 in green. The red line indicates the year 2017, marking the point when the composition of the PEDV non-indel strain population changes, and several ancestral clades have not been detected since.

**Figure 6 viruses-17-00654-f006:**
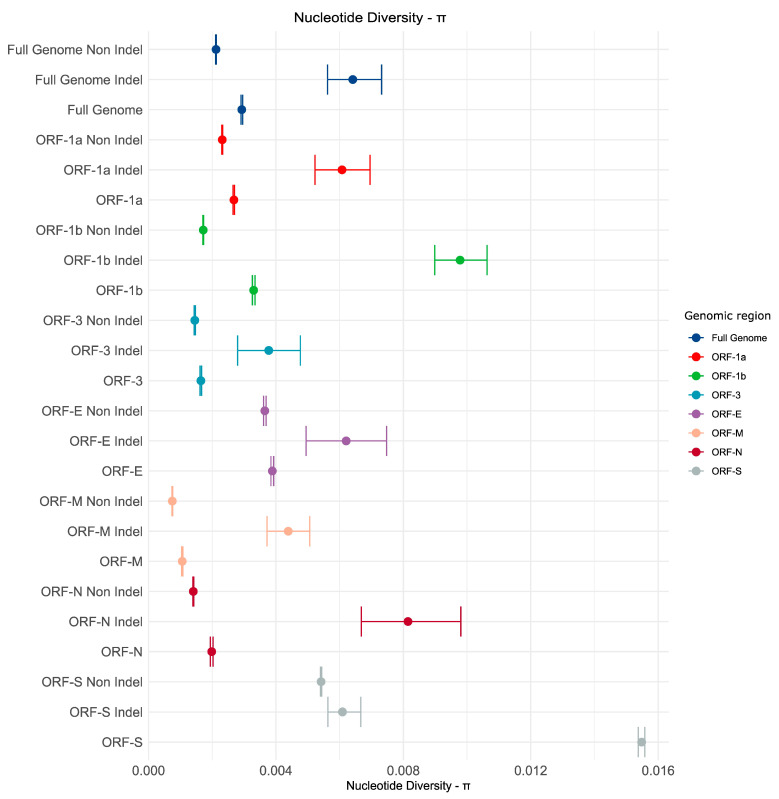
Genetic variability based on complete genomes of PEDV. Nucleotide diversity (π) values were estimated for the complete dataset and for each strain individually. We also estimated genetic diversity for each of the ORFs and for the complete dataset. In the graph, points represent the mean, and bars represent the 95% confidence intervals estimated through the bootstrap test.

**Figure 7 viruses-17-00654-f007:**
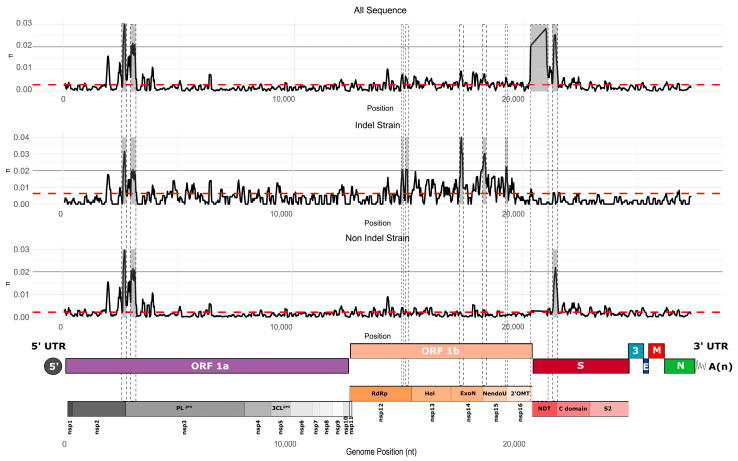
Average nucleotide diversity per site across the full genome, computed using a sliding window of 100 nucleotides with steps of 25 nucleotides. The red dotted line represents the overall mean nucleotide diversity. Nucleotide diversity was calculated for each strain individually as well as for the complete dataset. At the bottom of the graph, the genomic organization map of PEDV is shown. The gray backgrounds indicate hypervariable regions.

**Table 1 viruses-17-00654-t001:** Summary of recombination events detected in the PEDV. A. The analysis was performed, encompassing the entire coding region of the PEDV genome. B. The analysis was performed for PEDV Spike. Only recombination events detected by at least four different methods and with *p*-values lower than a Bonferroni-corrected α = 0.05 were considered significant. # Abbreviations of detection methods: R, Rdp; G, Geneconv; B, Boostcan; M, Maxichi; C, Chimaera; S, Siscan; 3, 3Seq. * Lowest *p*-value reported by the method in bold and underline.

		**Breakpoint Position**						
**A**								
**Full Genome**		**In Recombinant Sequence**						
**Event**	**Found In**	**Begin**	**End**	**Recombination** **Sequencies**	**Minor Parental**	**Major Parental**	**Detection Methods #**	***p*-Value ***
1	6	16,816	22,432	KR265760	KM975738	KF452322	RGBMCSP**3**	2.64 × 10^−76^
2	5	3910	20,375	KR265761	KR265763	KM975738	RGMCS**3**	6.01 × 10^−75^
3	4	13,244	21,566	KM975740	KM975738	KF452322	RGMS**3**	3.03 × 10^−53^
4	2	13,244	17,879	KR265786	KJ645704	KF468753	RGMC**3**	3.02 × 10^−46^
5	2	11,075	27,425	KM975738	KR265759	Unknown	**RGBMCS3**	5.96 × 10^−37^
6	106	16,816	19,902	KU558702	KJ645704	248	RGMC**3**	3.48 × 10^−32^
7	12	20,376	21,478	KR265761	Unknown	KJ645643	RGBM**S**3	1.48 × 10^−33^
8	7	4545	16,815	KR265759	Unknown	205	RGBMS**3**	7.98 × 10^−28^
9	6	4377	16,815	264	Unknown	KJ645641	MCS**3**	1.16 × 10^−22^
10	116	9921	14,874	KU893873	KF452322	KJ645635	RGMC**3**	3.91 × 10^−19^
11	223	7680	13,243	KJ645704	KJ645635	KR265844	RGM**3**	1.23 × 10^−14^
12	152	19,912	21,587	KJ645641	MG837058	Unknown	RGMC**3**	1.20 × 10^−6^
**B**								
**Spike Protein**		**Breakpoint Position**						
**Event**	**Found In**	**Begin**	**End**	**Recombination** **Sequencies**	**Minor Parental**	**Major Parental**	**Detection Methods #**	***p*-Value ***
1	3	8	915	KU982979	265	340	RM**S**P3	1.75 × 10^−23^
2	34	8	955	340	Unknown	296	RBM**S**P3	4.98 × 10^−24^
3	24	8	1056	262	Unknown	KU982968	M**P**3	3.88 × 10^−23^

**Table 2 viruses-17-00654-t002:** Pairwise nucleotide identity computed for the spike protein coding region, values were estimated using SDT.v1.2. Genetic distances were computed within and between clades. The table indicates the highest, lowest, and average values, respectively.

Clade	Min	Max	Average
Spike Protein			
Non-S-INDEL	97.6%	99.8%	99.1%
Clade1^non-S-INDEL^	98.2%	99.8%	99.3%
Clade1^non-S-INDEL^ vs. older clades^non-S-INDEL^	97.7%	99.6%	98.6%
Clade2^non-S-INDEL^	99.3%	99.9%	99.2%
Clade2^non-S-INDEL^ vs. older clades^non-S-INDEL^	97.8%	99.7%	99.3%
Clade1^non-S-INDEL^ vs. Clade 2^non-S-INDEL^	97.9%	99.0%	98.6%
S-INDEL strain	94.5%	100.0%	99.1%
S-INDEL vs. non-S-INDEL strains	91.14%	94.25%	93.61%
Full Genome			
Non-S-INDEL strain	98.75%	100%	99.78%
INDEL strain	99.21%	100%	99.55%
INDEL vs. Non-INDEL strains	98.52%	99.85%	98.99%

**Table 3 viruses-17-00654-t003:** Nonsynonymous to synonymous substitution ratios (dN/dS) and selection analysis of PEDV structural proteins. The table presents the dN/dS ratios calculated using the SLAC method and the number of negatively and positively selected sites identified using three distinct methods: SLAC, MEME, and FUBAR. These analyses were performed for each structural protein of PEDV. A dN/dS ratio < 1 suggests purifying selection, a dN/dS ratio = 1 indicates neutral evolution and a dN/dS ratio > 1 suggests positive selection.

Gene	dN/dS	SLAC		MEME		FUBAR	
		Diversifying	Purifying	Diversifying	Purifying	Diversifying	Purifying
**S-INDEL strain**							
Spike	0.331	-	368, 659	27, 83, 240, 351, 429, 500, 632	-	83, 196, 310, 351, 719	21, 41, 44, 48, 73, 76, 92, 93, 125, 141, 149, 199, 226, 237, 238, 269, 312
Envelop	0.491	-	-	-	-	66	-
Matrix	0.0967	-	-	-	-	-	27, 41, 71, 78, 116, 121, 122, 188, 201
Nucleo capside	0.240	-	-	27, 54, 415	-	27	28, 43, 51, 140, 147, 207, 211, 249, 267, 271, 298, 327, 360, 364, 414
**Non-S-INDEL**							
Spike	0.887	144, 380, 488, 525, 526, 568, 610, 614, 724	80, 94, 100, 101, 114, 139, 154, 209, 276, 359, 374, 462, 558, 582, 588, 621, 654, 664, 697, 729	24, 58, 144, 277, 355, 380, 412, 417, 433, 488, 495, 525, 526, 568, 610, 614, 724	-	24, 55, 58, 70, 144, 157, 196, 277, 355, 380, 412, 433, 488, 494, 501, 525, 526, 568, 610, 614, 637, 653, 676, 695, 724	53, 94, 100, 101, 109, 139, 154, 244, 276, 359, 374, 394, 462, 468, 558, 582, 621, 625, 654, 697, 729
Envelop	0.743	-	-	-	-	-	-
Matrix	0.195	-	** - **	-	200	-	-
Nucleo capside	0.585	-	240	-	-	193	55, 87, 100, 190, 240, 244, 252, 386

## Data Availability

Portions of the dataset are privately owned by the production systems and may be subject to restrictions. However, the data can be made available upon reasonable request to the corresponding author, contingent upon permission from the respective production systems. Publicly available sequences from GenBank can be accessed and downloaded in the following link https://github.com/herrerasilva-evol-viralland/Current-evolutionary-dynamics-of-Porcine-Epidemic-Diarrhea-Virus-PEDV-in-the-U.S.- (accessed on 27 April 2025).

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
