# Peer review of "Current Evolutionary Dynamics of Porcine Epidemic Diarrhea Virus (PEDV) in the U.S. a Decade After Introduction"

_viruses, 2025, doi:10.3390/v17050654_

Round 1
Reviewer 1 Report
Comments and Suggestions for Authors
In the manuscript entitled "Current evolutionary dynamics of Porcine Epidemic Diarrhea Virus (PEDV) in the U.S. a decade after introduction", the authors analyzed 313 genomes and 556 Spike protein sequences including non-S-INDEL strain and S-INDEL strain, found that non-S-INDEL strain is the predominant in prevalence. S-IDEL occurred less recombination inter-strain and non-S-IDEL sequences on the opposite and are further divided into three smaller clusters. The paper also characterized genetic diversity and variability of PEDV, revealing that the hypervariable region in nsp2 and nsp3. S-IDEL strain also showed S hypervariable region in the NTD. These data will lead us understanding the evolution of PEDV.
Specific comments:
- Please describe more results of diversity and variability, and highlight the meaning of the results.
- Please complete the keywords parts.
- Line 276, as shown in Figure 3, non-S-INDEL cluster was subdivided into three groups, not four groups.
- Line 271 to 275, the describe about S-INDEL and non-S-INDEL is so confusing, please clarify it.
- Line 379 to 380, based on the observation of figure 7, both S-INDEL and non-S-INDEL exhibited hypervariable regions in the nsp2 and nsp3 regions, not nsp1. Please clarify this point.

Overall, the paper requires grammatical and logical overhaul.
- Line 17, parentheses is missing after “PEDV”.
- Line 33, “between-farm spread” should be “between farm-spread”
- Line 42 “coronaviridae” should be “coronaviridae”, which is italicized.
- Line 45, “a structural protein” should be “structural proteins”.
- Line 146, “,”shoud be deleted.
- Line 153, “values” the first letter should be uppercase.
Author Response
Current evolutionary dynamics of Porcine Epidemic Diarrhea Virus (PEDV) in the U.S. a decade after introduction
Joao P. Herrera da Silva1, Nakarin Pamornchainavakul1, Mariana Kikuti1, Xiaomei Yue1, Cesar A. Corzo1 and Kimberly VanderWaal1.
|
Response to Reviewer X Comments
|
||
|
1. Summary |
|
|
|
We would like to thank the reviewers for taking the time to review this manuscript. Their feedback has been valuable in improving this manuscript. Below, I have copied the reviewers’ comments in full, and a point by point response is given in Red.
|
||
Comments 1:
Please describe more results of diversity and variability, and highlight the meaning of the results.
Response 1:
Thank you for pointing this out. We have modified the text to make it more descriptive and have also added an interpretation of the results. Below, you will find the modifications. Please let us know if any further changes are needed. The modification can be found in the new version.
lines 334 to 338
…Genetic variability was assessed for the complete set of sequences and for each strain individually. Nucleotide diversity was estimated for complete genomes. We also estimated the genetic diversity for each individual coding regions to evaluate the relative contribution of the genetic diversity of each gene to the diversity observed in the complete genome (Figure 6)…
Lines 357 to 363
…The differences in nucleotide diversity between genes indicate that each region of the genome exhibits with some genomic regions being more prone to accumulating genetic variation than others. The hypervariable regions in ORF-S and ORF-1b appear to coincide with regions where recombination has been detected, suggesting a possible explanation for the high genetic variation. We also observed differences between the two strains regarding which genes show greater genetic diversity, suggesting distinct evolutionary trajectories between the two PEDV strains.
Comments 2:
Please complete the keywords parts.
Response 2:
Thank you for noticing that, for some reason, it went unnoticed. The keywords are included; you can find them on line 24.
“Keywords: PEDV; Coronavirus; Recombination; Genetic diversity; temporal dynamics ”
Comments 3:
Line 276, as shown in Figure 3, non-S-INDEL cluster was subdivided into three groups, not four groups
Response 3:
Thank you for pointing this out. Figure 3 shows four clusters, not three there may have been a misunderstanding. The figure that contains three non-IDEL clusters is Figure 2, which refers to the phylogenetic networks based on complete genomes. Please take another look
Comments 4:
Line 271 to 275, the describe about S-INDEL and non-S-INDEL is so confusing, please clarify it.
Response 4:
Thank you very much for your comment. We have modified the text to make it as clear as possible. If this is still not sufficient, please let us know and provide a bit more detail about what remains unclear. You can find the modified text in Line 260 to 269, and also you can see the new text below.
“A long branch separating the S-INDEL from the non-S-INDEL strains indicates a high level of divergence between them. The high degree of reticulation was observed in the backbone of the phylogenetic network, particularly in the region where the S-INDEL and non-S-INDEL strains diverge, suggesting occurrences of inter-strain recombination. A lower degree of reticulation was observed along the internal branches of the S-INDEL strain, suggesting that intra-strain recombination for S-INDEL was less frequent (Figure 3). The PEDV non-S-INDEL cluster was subdivided into four groups, a high degree of reticulation was observed, suggesting that intra-strain recombination events were more frequent among non-INDEL strains (Figure 3). The recombination evidence was confirmed by the PHI test (p < 0.001).”
Comments 5:
Line 379 to 380, based on the observation of figure 7, both S-INDEL and non-S-INDEL exhibited hypervariable regions in the nsp2 and nsp3 regions, not nsp1. Please clarify this point.
Response 5:
I sincerely appreciate your attention to this error. We have revised the text accordingly. The modified version can be found below, and you can also find it in the final version, from line 345 to 347.
“Both exhibited hypervariable regions in the 5′ end of the genome, specifically within the nsp2 and nsp3 regions.”
Comments 6:
Minor comments
Line 17, parentheses is missing after “PEDV”.
Line 33, “between-farm spread” should be “between farm-spread”
Line 42 “coronaviridae” should be “coronaviridae”, which is italicized.
Line 45, “a structural protein” should be “structural proteins”.
Line 146, “,”shoud be deleted.
Line 153, “values” the first letter should be uppercase.
Response 6:
Thank you for your attention to all of this detail. We have implemented all the minor comments except for the one on line 33, as we are confident that 'between-farm spread' is correct. In this context, 'between-farm' functions as a compound adjective describing the type of spread, specifically one that occurs between farms. As such, the use of a hyphen is grammatically appropriate. The alternative phrasing, 'between farm-spread', would not be correct in standard English usage.

Reviewer 2 Report
Comments and Suggestions for Authors
This study investigates the evolutionary dynamics of Porcine Epidemic Diarrhea Virus (PEDV) in the U.S. over a decade post-introduction (2013–2024). By analyzing 313 genomes and 556 Spike protein sequences, the authors found that the non-S-INDEL strain (G2b) has become dominant, with two new, geographically localized sub-clades emerging after 2017. Recombination events, particularly in ORF1a/b and Spike, drive diversification, and purifying selection shapes structural proteins. Genetic diversity is higher in non-S-INDEL, but S-INDEL exhibits elevated diversity in ORF1b/N. The study highlights limited inter-regional spread and underscores the need for sustained genomic surveillance to confirm clade extinction and track viral evolution. However, I have some concerns.
Major concern:
The conclusion that ancestral clades are extinct relies heavily on sparse post-2017 sequencing data. While plausible, insufficient sampling in endemic phases (e.g., <10 sequences/year after 2019) weakens this claim.
Minor concern:
(1) Figure 1: The labeling on the y-axis, described as "cases/sequences," is ambiguous. Please clarify whether "cases" refers to outbreaks, farms, or herds. Furthermore, the x-axis should display all continuous years to accurately represent the data over time.
(2) Figures 2 and 3: Legends should define reticulations and bootstrap values. Clarify how clusters were assigned (e.g., genetic distance thresholds).
(3) Figure 7, Label hypervariable regions (e.g., nsp3, S1-C-domain) directly on the plot for clarity.
(4) Please provide the original FASTA files as supplemental data used for the analysis of the S protein. This information is crucial for the assessment and reproducibility of the study.
(5) Data Sources: Clarify how GenBank and MSHMP sequences were selected (random vs. convenience sampling). Address potential biases (e.g., overrepresentation of certain states/production systems).
(6) Temporal Signal Analysis: Specify clock model selection criteria (e.g., Bayes factors) in BETS.
(7) In discussion, the authors should acknowledge potential biases from uneven sampling across regions/production systems.
Comments on the Quality of English Language
Minor grammatical errors exist ("evidences" → "evidence"). Simplify complex sentences (e.g., "Our results suggest non-S-INDEL strain is..." → "Our results suggest the non-S-INDEL strain is...").
Author Response
Current evolutionary dynamics of Porcine Epidemic Diarrhea Virus (PEDV) in the U.S. a decade after introduction
Joao P. Herrera da Silva1, Nakarin Pamornchainavakul1, Mariana Kikuti1, Xiaomei Yue1, Cesar A. Corzo1 and Kimberly VanderWaal1.
|
Response to Reviewer X Comments
|
||
|
1. Summary |
|
|
|
We would like to thank the reviewers for taking the time to review this manuscript. Their feedback has been valuable in improving this manuscript. Below, I have copied the reviewers’ comments in full, and a point by point response is given in Red.
|
||
Reviewer 2
|
Comments 1: The conclusion that ancestral clades are extinct relies heavily on sparse post-2017 sequencing data. While plausible, insufficient sampling in endemic phases (e.g., <10 sequences/year after 2019) weakens this claim. |
|
Response 1: Thank you very much for your comment. It was a very valuable observation, and we agree with your point, indeed, our dataset after 2017 is quite limited. We have revised the text to present this as an open hypothesis that needs to be confirmed. The modification can be found in the new version. Please let us know if there is still a need to further soften this statement. Please check line 466 and 532-533 Our phylogenetic analyses based on the S1 subunit of the Spike protein suggest that many of the early clades circulating during the epidemic period are potentially no longer be circulating in the U.S. at present. Many of older clades appear to have left no descendants. Based on the available data, nearly all sequences sampled after 2017 clustered into two current clades. Lines 527 to 528
|
|
Comments 2: 1) Figure 1: The labeling on the y-axis, described as "cases/sequences," is ambiguous. Please clarify whether "cases" refers to outbreaks, farms, or herds. Furthermore, the x-axis should display all continuous years to accurately represent the data over time. |
Response 2:
Thank you very much for your comment and attention to detail. For some reason, we had not included the labels in the graph. We have implemented the modification, and you can find it in the final version as well as in the attached file below.
|
Comments 3: Figures 2 and 3: Legends should define reticulations and bootstrap values. Clarify how clusters were assigned (e.g., genetic distance thresholds).
|
|
Response 3: We sincerely appreciate the feedback; however, this concept is explained in the text (reticulations are non-bifurcating patterns observed in the network, which indicate the possibility of sharing more than one common ancestor). Regarding bootstrap values, by default they are not typically reported for this type of visualization, and for aesthetic reasons, displaying them in dense networks with long branch lengths such as those presented here would be virtually impossible to visualize. What is crucial in this case, and supports the evidence of recombination, are the results from the PHI test. The criterion for defining clusters was based on the network topology, and we have incorporated this explanation throughout the text.
|
|
Comments 4: Figure 7, Label hypervariable regions (e.g., nsp3, S1-C-domain) directly on the plot for clarity. |
|
Response 4: Thank you very much for your observation; it was truly helpful in improving the figure. This way, it becomes more obvious for the reader to visualize the regions we are referring to. We have implemented the modification, and you can find it both in the text and below this comment.
|
|
Comments 5: Please provide the original FASTA files as supplemental data used for the analysis of the S protein. This information is crucial for the assessment and reproducibility of the study.
|
|
Response 5: Thank you for pointing this out. We agree with this comment. Unfortunately, due to data sensitivity, we did not have authorization to deposit all the sequences we analyzed in this study. However, we were granted permission to deposit some of them in GenBank, including several complete genomes and some spike protein sequences representing the clades currently circulating in the USA. Two data subsets can be found at the link below. These subsets include all sequences downloaded from GenBank as well as some sequences made available by the MSHMP. We hope this is sufficient. https://github.com/herrerasilva-evol-viralland/Current-evolutionary-dynamics-of-Porcine-Epidemic-Diarrhea-Virus-PEDV-in-the-U.S.- |
|
Comments 6: Data Sources: Clarify how GenBank and MSHMP sequences were selected (random vs. convenience sampling). Address potential biases (e.g., overrepresentation of certain states/production systems). |
|
Response 6: Thank you for your comment. Actually, we did not perform any subsampling process; we used all available sequences in both datasets (Lines 89-90). We have implemented your suggestion. Throughout the discussion, we mentioned the potential biases in our datasets. In addition, we added an extra paragraph to address the study’s limitations (Lines 641 to 647). Lines 89-90 Lines 641 to 647 Our sequence dataset covers 21 different states, encompassing four major swine-producing regions: the Northeast, Midwest, Great Plains, and Atlantic Seaboard. Approximately 263 sequences lacked information regarding their geographic origin. Overall, the temporal distribution of sequences from each of the four regions was relatively similar, except for the epidemic period, during which some areas contributed a higher number of sequences. However, we were not able to assess if sequencing effort was equal across production systems within regions, which could mean that some genetic diversity was missed.
Comments 7 (6) Temporal Signal Analysis: Specify clock model selection criteria (e.g., Bayes factors) in BETS. Response 7: The choice of molecular clock was also based on the log Bayes factor. Thank you for pointing that out. We have added a sentence specifying the criterion used (Lines 311 to 313).
Comments 8 In discussion, the authors should acknowledge potential biases from uneven sampling across regions/production systems. Response 8: We added a paragraph to acknowledge the limitations, which you can find in the lines 441 to 447
Lines 441 to 447 Our sequence dataset covers 21 different states, encompassing four major swine-producing regions: the Northeast, Midwest, Great Plains, and Atlantic Seaboard. Approximately 263 sequences lacked information regarding their geographic origin. Overall, the temporal distribution of sequences from each of the four regions was relatively similar, except for the epidemic period, during which some areas contributed a higher number of sequences. However, we were not able to assess if sequencing effort was equal across production systems within regions, which could mean that some genetic diversity was missed.
Comments 9 Minor grammatical errors exist ("evidences" → "evidence"). Simplify complex sentences (e.g., "Our results suggest non-S-INDEL strain is..." → "Our results suggest the non-S-INDEL strain is..."). Response 9: We appreciate your effort in identifying these minor errors that had gone unnoticed by us. We have made the suggested modifications.
|

Reviewer 3 Report
Comments and Suggestions for Authors
The authors have analysed several hundreds of PEDV sequences circulating in the US during the last 12 years. More than 500 spike protein sequences and additional 313 whole PEDV genomes were screened for recombination events and the phylogenetic trees were constructed. The methods are described quite in details; seven figures and three tables are added.
The manuscript is well arranged and written in a clear manner.
I have just some minor comments:
The numeration of the pages should be checked carefully as the numbers do not tally with the reality. There are 36 pages in the manuscript, however, the last page has number 11.
P2, L45: "...four ORFs that encodes for..." Please, correct the incorrect plural ("...four ORFs that encode for...").
P22(?), L380: "...specifically within the nsp1 and nsp3 regions." Please, correct the typo (should be "...specifically within the nsp2 and nsp3 regions.").
Author Response
Current evolutionary dynamics of Porcine Epidemic Diarrhea Virus (PEDV) in the U.S. a decade after introduction
Joao P. Herrera da Silva1, Nakarin Pamornchainavakul1, Mariana Kikuti1, Xiaomei Yue1, Cesar A. Corzo1 and Kimberly VanderWaal1.
|
Response to Reviewer X Comments
|
||
|
1. Summary |
|
|
|
We would like to thank the reviewers for taking the time to review this manuscript. Their feedback has been valuable in improving this manuscript. Below, I have copied the reviewers’ comments in full, and a point by point response is given in Red.
|
||
Reviewer 3
|
Comments 1: The numeration of the pages should be checked carefully as the numbers do not tally with the reality. There are 36 pages in the manuscript, however, the last page has number 11. |
|
Response 1: Thank you for that. We have incorporated the correct numbering, you can verify it in the revised document.
|
|
Comments 2: P2, L45: "...four ORFs that encodes for..." Please, correct the incorrect plural ("...four ORFs that encode for..."). |
Response 2: Thank you for your attention to detail, we truly appreciate it. For some reason, these small mistakes went unnoticed by us. We have made the necessary changes, which you can find in line 48 to 51.
line 48 to 51
“…Its genome consists of ORF1a and ORF1b, encoding 16 non-structural proteins involved in replication, transcription, and immune evasion, and other four ORFs that encode for structural protein: the spike protein (S), nucleocapsid (N), membrane (M), envelope (E), as well as the accessory protein ORF3….”
|
Comments 3: P22(?), L380: "...specifically within the nsp1 and nsp3 regions." Please, correct the typo (should be "...specifically within the nsp2 and nsp3 regions."). |
|
Response 3: We greatly appreciate you noticing this error. We have made the necessary modifications, which you can find in the text (line 345 to 347) as well as below this comment
line 345 to 347.
|

Reviewer 4 Report
Comments and Suggestions for Authors
References are in parentheses; they should be enclosed in brackets [] to differentiate the information.
- Figure 1 lacks the legends for the red and blue lines (even though they are described in the figure caption), and the legend "case numbers" should be added to the vertical axis.
- Lines 207 to 215 state that some of the recombinant events were detected in n sequences analyzed. In Table 1A, if you add up the supposed times a recombinant event was found, it results in 641, which would indicate that technically all the sequences analyzed (n=566) are results of recombination.
I think there is an error. Explain how you determined this and attach the RDP5 outputs. The question is: Did you compare the breakpoints of each detected recombination event, and if they occur at the same position within the genome, to determine how many recombination events it occurs in? The above would be a strong indication that it is the same event.
Author Response
Current evolutionary dynamics of Porcine Epidemic Diarrhea Virus (PEDV) in the U.S. a decade after introduction
Joao P. Herrera da Silva1, Nakarin Pamornchainavakul1, Mariana Kikuti1, Xiaomei Yue1, Cesar A. Corzo1 and Kimberly VanderWaal1.
|
Response to Reviewer X Comments
|
||
|
1. Summary |
|
|
|
We would like to thank the reviewers for taking the time to review this manuscript. Their feedback has been valuable in improving this manuscript. Below, I have copied the reviewers’ comments in full, and a point by point response is given in Red.
|
||
Reviewer 4
|
Comments 1: Figure 1 lacks the legends for the red and blue lines (even though they are described in the figure caption), and the legend "case numbers" should be added to the vertical axis. |
|
Response 1: Thank you very much for pointing this out. For some reason, this detail went unnoticed by us. We have made the modification in the figure, which can be found in the final version of the text as well as below this comment.
|
|
Comments 2: Lines 207 to 215 state that some of the recombinant events were detected in n sequences analyzed. In Table 1A, if you add up the supposed times a recombinant event was found, it results in 641, which would indicate that technically all the sequences analyzed (n=566) are results of recombination. I think there is an error. Explain how you determined this and attach the RDP5 outputs. The question is: Did you compare the breakpoints of each detected recombination event, and if they occur at the same position within the genome, to determine how many recombination events it occurs in? The above would be a strong indication that it is the same event
|
|
Response 2: Thank you for your comment. However, I believe there may have been a misunderstanding regarding the interpretation of the results presented in Table 4A. This table refers specifically to whole-genome sequence analyses, comprising a total of 313 sequences. It seems that the values in the “Found in” column may have been summed to estimate the number of recombination events. However, this approach does not accurately reflect the number of sequences analyzed. It’s important to note that a single sequence can exhibit more than one independent recombination event in different genomic regions. Therefore, summing these values can lead to this type of misinterpretation, resulting in a total that exceeds the actual number of sequences analyzed. Additionally, the breakpoint start and end positions must be considered for a proper interpretation of the results. These details are clearly presented in the table to help prevent any confusion. Please also keep in mind that these genomes are over 28 kb in length, which allows for multiple distinct recombination events to occur within the same sequence. You can find the RDP5 output at the link below. I hope this helps clarify your question. https://github.com/herrerasilva-evol-viralland/Current-evolutionary-dynamics-of-Porcine-Epidemic-Diarrhea-Virus-PEDV-in-the-U.S.-
|

Round 2
Reviewer 4 Report
Comments and Suggestions for Authors
Thank you for your responses, the manuscript has improved appropriately.